# Pupillary response is associated with the reset and switching of functional brain networks during salience processing

**Hengda He** [1]☺*, **Linbi Hong** [1]☺, **Paul Sajda** [1,2,3,4]*

**1** Department of Biomedical Engineering, Columbia University, New York, New York, United States of America, **2** Department of Electrical Engineering, Columbia University, New York, New York, United States of America, **3** Department of Radiology, Columbia University, New York, New York, United States of America, **4** Data Science Institute, Columbia University, New York, New York, United States of America

☺ These authors contributed equally to this work.
* hengda.he@columbia.edu (HH); psajda@columbia.edu (PS)

## Abstract

The interface between processing internal goals and salient events in the environment involves various top-down processes. Previous studies have identified multiple brain areas for salience processing, including the salience network (SN), dorsal attention network, and the locus coeruleus-norepinephrine (LC-NE) system. However, interactions among these systems in salience processing remain unclear. Here, we simultaneously recorded pupillometry, EEG, and fMRI during an auditory oddball paradigm. The analyses of EEG and fMRI data uncovered spatiotemporally organized target-associated neural correlates. By modeling the target-modulated effective connectivity, we found that the target-evoked pupillary response is associated with the network directional couplings from late to early subsystems in the trial, as well as the network switching initiated by the SN. These findings indicate that the SN might cooperate with the pupil-indexed LC-NE system in the reset and switching of cortical networks, and shed light on their implications in various cognitive processes and neurological diseases.

**Data Availability Statement:** All data are in the manuscript and its supporting information files. Analysis scripts are available at: https://github.com/hehengda/EffectiveConnectivity_paper_code. Dataset and Demo are available at:

## Author summary

In a perceptual experience, the brain is constantly processing internal goals and salient events in the environment. It is thought that this processing recruits distinct brain networks, with the salience network (SN) having a central role. Here, by using a multimodal neuroimaging approach, our study identifies spatiotemporally dissociable brain networks, reflecting different cognitive processes in salience processing. Using the changes in pupil diameter as a proxy for the neural activity in the locus coeruleus-norepinephrine (LC-NE) neuromodulatory system, we found evidence that the SN and LC-NE system interact to assist the reorganization between internal and external cognitive processes in salience processing.

https://doi.org/10.6084/m9.figshare.22518010.v1.
The effective connectivity state-space model is
available at https://github.com/taotu/VBLDS_
Connectivity_EEG_fMRI.

**Funding:** The study was funded by a Vannevar
Bush Faculty Fellowship from the U.S. Department
of Defense (N00014-20-1-2027) and a Center of
Excellence grant from the Air Force Office of
Scientific Research (FA9550-22-1-0337), both to
P.S. The funders had no role in study design, data
collection and analysis, decision to publish, or
preparation of the manuscript.

**Competing interests:** I have read the journal's
policy and the authors of this manuscript have the
following competing interests: P.S. is a scientific
advisor to Optios Inc. and OpenBCI LLC. All other
authors declare they have no competing interests.

## Introduction

To navigate complex and dynamic environments our brains cannot allocate attention to everything, but instead must continuously mark and process salient objects [1]. For example, when we are walking on a busy street, we will likely direct attention to the traffic lights, a siren, and our planned route. In psychology and neuroscience, the term 'salience' refers to a noticeable or important object that stands out from the surroundings or background. Salience is usually accompanied by unexpectedness, novelty, and infrequency [2]. Typically, salience processing involves two general mechanisms [3]: 1) bottom-up processing that includes filtering and amplifying the sensory information; 2) top-down processing in support of anticipation, cognitive control, and goal-directed behaviors. To investigate salience processing, one of the widely used experimental paradigms is the oddball task, where subjects are instructed to detect distinct infrequent target stimuli in a stream of standard stimuli. In previous functional magnetic resonance imaging (fMRI) studies, a variety of brain areas have been identified as correlates of salience processing, including regions in the dorsal attention network (DAN), salience network (SN), sensory cortex, primary somatosensory cortex (S1), and subcortex [2, 4–6]. However, it is challenging to dissociate and interpret the distinct cognitive processes underlying these spatially distributed regions. Even though functional connectivity analyses have been used to dissociate brain networks [7], the lack of time scales and the directionality in the couplings of these brain regions and networks still hinder the inference of their roles in salience processing.

Besides cortical networks, the locus coeruleus (LC), as the primary source of norepinephrine (NE), has also been associated with salience processing. The phasic LC activity has been shown to produce the P300 event-related potential (ERP), which typically appears robustly following target stimuli (responds weaker following standard stimuli) in oddball paradigms [8, 9]. Besides the P300 ERP, pupil diameter has also been used as a psychophysiological marker of the LC activity [10]. For example, in a single-unit recording study, both the spiking activity in the LC and pupil diameter are evoked following unexpected auditory stimuli [11]. Trial-by-trial associations were also observed between the pupillary response magnitude and LC responses. The association between the activity in the LC and pupil diameter fluctuations has also been shown in an fMRI study with oddball paradigm [12]. Together, these findings indicate the reliability of LC-pupil relationships during neural processes of salient stimuli in the oddball paradigm. Pupil diameter fluctuations reflect salience, attention, surprise, efforts, and arousal [13]. In the oddball paradigm, target-driven pupil dilation reflects not only bottom-up processes, but also top-down cognitive processes of decision-making and task demands [13].

Both the cortical network dynamics and the LC-NE system have been well characterized, such as the network switching model of the SN and DAN [3], and the network reset and the adaptive gain theory of the LC [8, 14, 15]. Even though the SN, DAN, and the LC-NE system have been closely related to each other [8, 16–18] and all centrally positioned in salience processing, it is still unclear what their integrative roles are in the cognitive processes of salient events. Hence, it would be valuable to investigate the cortico-subcortical associations between the cortical network dynamics and pupil-indexed neuromodulatory systems, such as the LC-NE system. Critically, a better understanding requires the assessment of the directional couplings between cortical networks, and their associations with the pupil measurements (brain-pupil relationships) in the context of salience processing.

Emerging evidence from the recent literature indicates that neuromodulatory systems, such as the LC-NE system, are important factors in shaping functional network connectivity, reorganization, and dynamics [19–22]. Thus, in this study, we explored this possibility using simultaneous recordings of pupillometry, electroencephalography (EEG), and fMRI in an oddball

paradigm. We first used a single-trial variability (STV) EEG-informed fMRI analysis, which allowed us to map the neural cascade underlying salience processing. Second, with the functional connectivity (FC) analyses of the fMRI data, we were able to map dissociable spatiotemporal functional network organizations of these neural correlates. Then, by leveraging the temporal dynamics of EEG, we further characterized the directional interactions between these regions with an effective connectivity state-space model [23]. Finally, we assessed brain-pupil relationships, which indicate the cortico-subcortical associations between the cortical network dynamics and the pupil-indexed LC-NE system. Specifically, we hypothesized that the pupil-indexed LC activity is associated with the effective connectivity of salience processing functional networks. Our results suggest that pupil-indexed LC-NE system and the SN share an integrative role in the reset and switching of functional brain networks in salience processing.

## Results

All subjects responded to the task correctly with an accuracy of 99.4% ± 0.1% (mean ± SD; SD, standard deviation) in detecting oddballs, and the response time (RT) is 403.5 ± 66.9 (mean ± SD) ms.

### Pupillometry analysis

We used a MRI-compatible eye-tracking camera to track pupil diameter fluctuations in parallel to simultaneous recordings of EEG and fMRI which measured brain activity. Pupil diameter data were then preprocessed and epoched (see S1 Text for details). The stimuli-locked grand average pupil diameter fluctuations are shown in S1 Fig. We observed a slow pupil dilation evoked by the target oddball stimuli peaking around 1.4 s after the stimulus. To quantify the pupil dilation elicited by the salient stimuli, we extracted the maximum percentage pupil diameter change within each trial as task-evoked pupillary response (TEPR).

### Single-trial EEG-informed fMRI analysis

To identify the neural correlates involved in the salience processing cascade, we performed a single-trial EEG-informed fMRI analysis. Briefly, this method extracts the EEG components discriminating target versus standard trials at different temporal windows spanning the trial, and these are used to map the temporally evolved brain activities that are correlates of salience processing using the fMRI data. The area under the receiver operating characteristic curve (AUC) was used to evaluate the performance of the single-trial discrimination, and the AUC value is above 0.75 between 200 to 700 ms. More details are in the Materials and Methods and S2 Fig. As a sanity check, traditional EEG stimulus-locked ERP and traditional fMRI analysis results are included in the S3 and S4 Figs. We observed the P300 component with a peak around 390 ms in the ERP analysis, and regions in the DAN, SN, visual and auditory cortex, S1, and subcortex were identified as significant clusters in the traditional fMRI analysis of oddball effects. In the EEG-informed fMRI analysis, from the resulting group-level whole-brain blood-oxygen-level-dependent (BOLD) activation maps, we identified significant clusters (p < 0.05, cluster corrected) at specific windows as shown in Fig 1A. These results revealed brain regions associated with salient stimuli processing: left superior parietal lobule (lSPL) (225 ms; positive cluster), left S1 (lS1) (250, 275, 350, and 375 ms; positive cluster), left orbito-frontal cortex (lOFC) (375 ms; negative cluster), left inferior parietal lobule (lIPL) (375 ms; negative cluster), left frontal operculum and temporal pole (600 ms; negative cluster), right primary motor cortex (rM1) (225 ms; positive cluster), right secondary visual cortex (rV2) (275 ms; positive cluster), right SPL (rSPL) (275 and 300 ms; positive cluster), right OFC (rOFC)

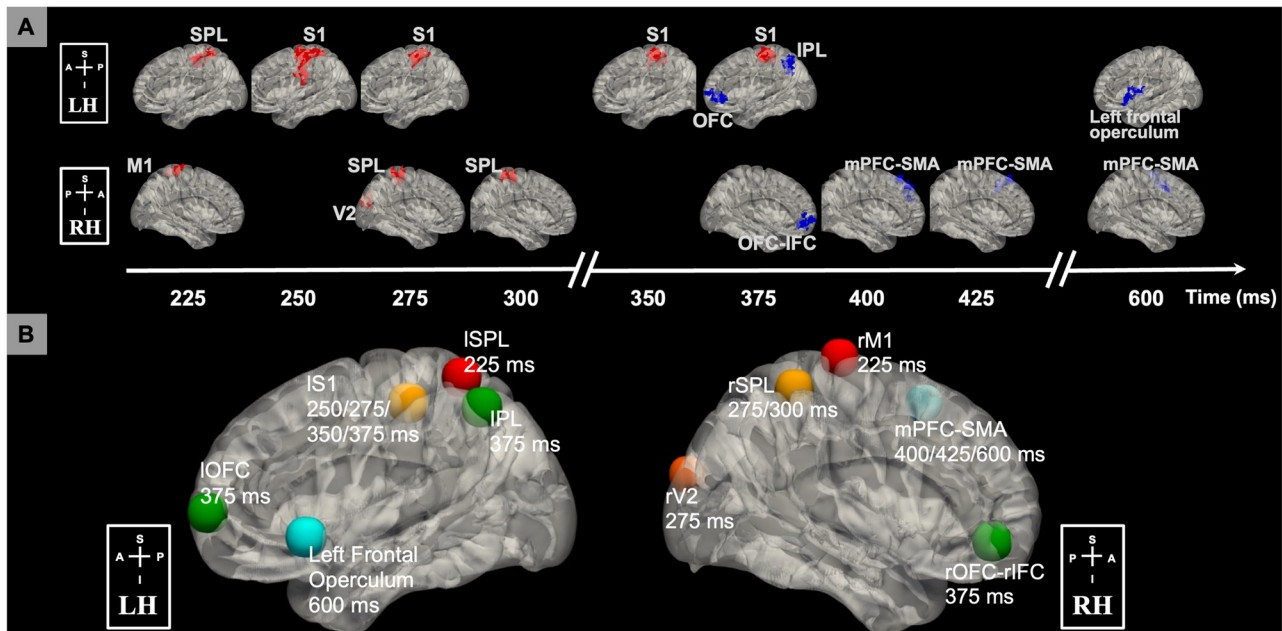

**Fig 1. Neural correlates of salience processing defined with the EEG single-trial variability (STV) informed fMRI analysis.** (A) Timing diagram showing significant group-level activation clusters (p < 0.05 cluster-wise multiple comparison correction). STV in EEG temporal components discriminating the target versus standard trials was used to map the spatiotemporally distributed BOLD fMRI correlates spanning the trial. EEG STV information was incorporated as BOLD predictors in voxel-wise general linear model (GLM) analysis of fMRI, controlling for the variance due to the presence of stimuli and response time (RT). Cluster colors denote positive (red) and negative (blue) effects. Time is relative to stimulus onset. (B) Definition of salience processing nodes. Each node is a sphere centered on the peak voxel of the group-level STV EEG-informed fMRI analysis results. Centroid of peak locations was used for regions involved in more than one temporal windows. Node colors denote timing of involvement in the trial from early to late (temporal order: red, orange, yellow, green, and blue). All clusters and nodes were overlaid on a 3D Montreal Neurological Institute (MNI) 152 brain pial surface for visualization. BOLD, blood-oxygen-level-dependent; RH, right hemisphere; LH, left hemisphere; A, anterior; P, posterior; S, superior; I inferior; SPL, superior parietal lobule; M1, primary motor cortex; S1, primary somatosensory cortex; V2, secondary visual cortex; OFC, orbitofrontal cortex; IPL, inferior parietal lobule; IFC, inferior frontal cortex; mPFC, medial prefrontal cortex; SMA, supplementary motor area.

and inferior frontal cortex (rIFC) (375 ms; negative cluster), and supplementary motor area (SMA) and medial prefrontal cortex (mPFC) (400, 425, and 600 ms; negative cluster). These results indicate a coordinated task-related neural cascade, representing the spatiotemporal dynamics of the neural correlates in salience processing.

## Network organization of brain regions associated with salience processing

Following this observed neural cascade associated with salience processing, a natural question we asked was about the organization of these spatiotemporally distributed regions. Specifically, we aimed to assess the network organization and connectivity between these brain regions. Thus, we defined 10 nodes: lSPL [x = -34, y = -52, z = 64; Montreal Neurological Institute (MNI) coordinates], lS1 (x = -46, y = -28, z = 52), lOFC (x = -40, y = 60, z = 4), lIPL (x = -48, y = -60, z = 50), left frontal operculum and temporal pole (x = -54, y = 16, z = -6), rM1 (x = 18, y = -22, z = 76), rV2 (x = 8, y = -94, z = 22), rSPL (x = 38, y = -42, z = 60), rOFC-rIFC (x = 42, y = 46, z = -8), mPFC-SMA (x = 4, y = 18, z = 56) as shown in Fig 1B (see S1 Text for details of salience processing node definition).

Given the emerging evidence that indicates the relevance between task activation and the intrinsic network organization of the brain [24], we hypothesized that the previously identified

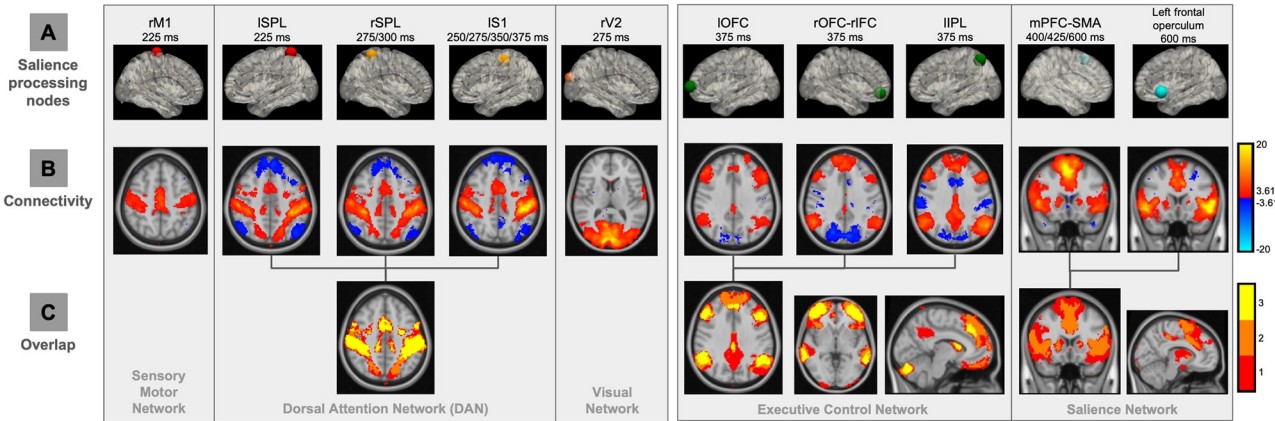

**Fig 2. Network localization approach to map functional networks underlying salience processing nodes.** (A) BOLD signals from the nodes (intersected with the gray matter mask) were extracted, controlling the nuisance signals (motion-related, ventricle, and white matter signals). (B) Group-level functional connectivity (FC) results of each node (t-value, mixed-effect, p < 0.001 uncorrected). Seed-based FC analysis (with the task-related variability regressed out) was used to map network of regions connected to each node location. Colors denote positive (red) and negative (blue) correlations. (C) Spatial overlaps in the FC maps of each node identified spatial network organizations of salience processing nodes. Colors represent the number of FC maps overlapped. lSPL and rSPL, left and right superior parietal lobule; rM1, right primary motor cortex; lS1, left primary somatosensory cortex; rV2, right secondary visual area; lOFC and rOFC, left and right orbitofrontal cortex; lIPL, left inferior parietal lobule; rIFC, right inferior frontal cortex.

nodes might represent organized underlying brain networks involved in salience processing. To test this hypothesis, a network localization approach was performed to map the brain regions functionally connected with each node (see Materials and methods for details; results in Fig 2). In the earliest time windows (225 to 275 ms), as expected, node rM1 was localized within the sensory motor network, and rV2 was part of the visual network. lSPL, rSPL, and lS1, which are spatiotemporally heterogeneous regions in the salience processing cascade, were mapped to a single brain network, i.e. DAN. Similarly, lOFC, rOFC-rIFC, and lIPL, which are all correlated with the EEG discriminating components at 375 ms post-stimulus, fell within the executive control network (ECN) [7]. Finally, in the latest time windows (400 to 600 ms), we found the nodes correlated with the late discriminating components, i.e. mPFC-SMA, left frontal operculum and temporal pole, were part of the SN. Overall, this observation suggests that these distributed nodes are spatially organized into intrinsic brain networks, indicating that the temporal evolution of different task activation regions (Fig 1A) spanning the trial might be supported by a specific set of brain networks (Fig 2).

Following these results that distinct nodes might fall within a common network, our next objective was to directly examine the functional connectivity between the nodes. As shown in Fig 3 (p < 0.05, uncorrected), in line with the previously observed spatial organization of the nodes revealed by network connectivity (Fig 2), we found strong connections across the nodes within each network. For example, lSPL, rSPL, and lS1 showed a stronger within network (i.e. nodes of DAN) connectivity compared to their connections with other nodes. Furthermore, the functional connectivity results clearly identified three distinct groups of the nodes, organized by the EEG discriminating component time windows, indicating a temporal organization of the nodes. Thus, to assess the brain networks involved in the task-related neural cascade, we defined three intrinsically connected salience processing networks, i.e. early-time (nodes: lSPL, rM1, rV2, rSPL, and lS1), middle-time (nodes: lIPL, lOFC, and rOFC-rIFC), and late-time (nodes: mPFC-SMA, left frontal operculum and temporal pole) networks.

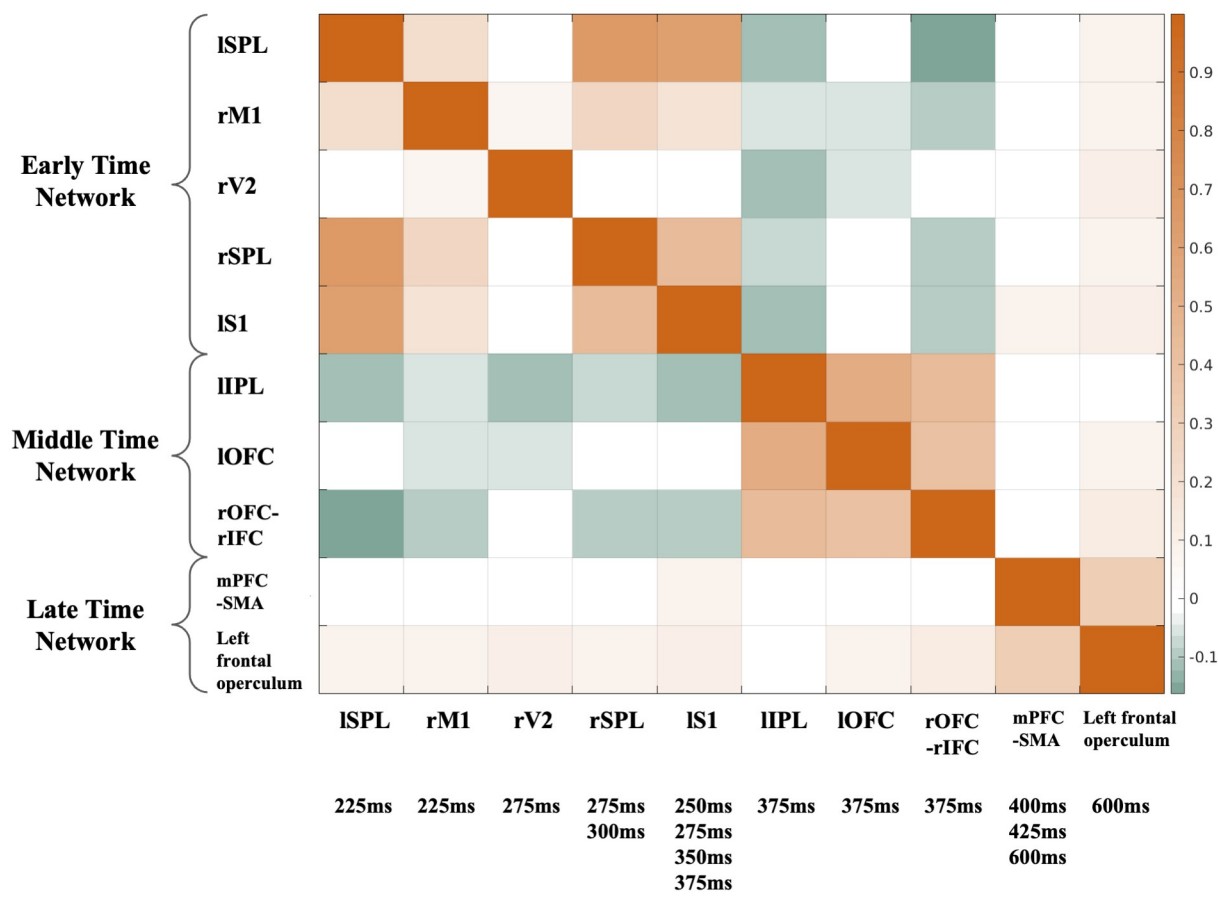

**Fig 3. Functional connectivity (FC) across salience processing nodes (group averaged z-score, mixed-effect, p < 0.05 uncorrected).** fMRI BOLD signals from the nodes (intersected with the gray matter mask) were extracted, controlling the nuisance signals (motion-related, ventricle and white matter signals). Pearson's correlation was calculated between BOLD signals from the nodes (with the task-related variability regressed out). FC results identified three distinct groups of the nodes, organized by the EEG discriminating component time windows, indicating a temporal network organization of the nodes: 1) early-time network includes lSPL and rSPL, rM1, rV2, and lS1; 2) middle-time network includes lOFC and rOFC, lIPL, and rIFC; 3) late-time network includes mPFC, SMA, left frontal operculum and temporal pole.

## Modulated effective connectivity by salience processing

In the functional connectivity analyses of fMRI data described above, we examined the spatial organizations of the salience processing nodes. Our next step was to investigate the temporal dependence and directional interaction between these nodes, by fitting EEG data with a state-space effective connectivity (EC) model (see Materials and methods for details). In the group-level analysis, results for the significant salient stimuli modulated EC (Bayesian parameter averaging; $\alpha < 0.05$; Bonferroni corrected) are shown in S5 Fig, with positive and negative connections shown in Fig 4. To quantify the connection strength of each node, we computed the total connection strength (S6 Fig), which is the sum of all the unsigned connection parameters (afferent, efferent, and self-connection) associated with the node. These results showed that the mPFC-SMA and lSPL have the highest afferent and efferent connection strength, respectively, suggesting that the mPFC-SMA and lSPL are the hubs in the processing of salient stimuli.

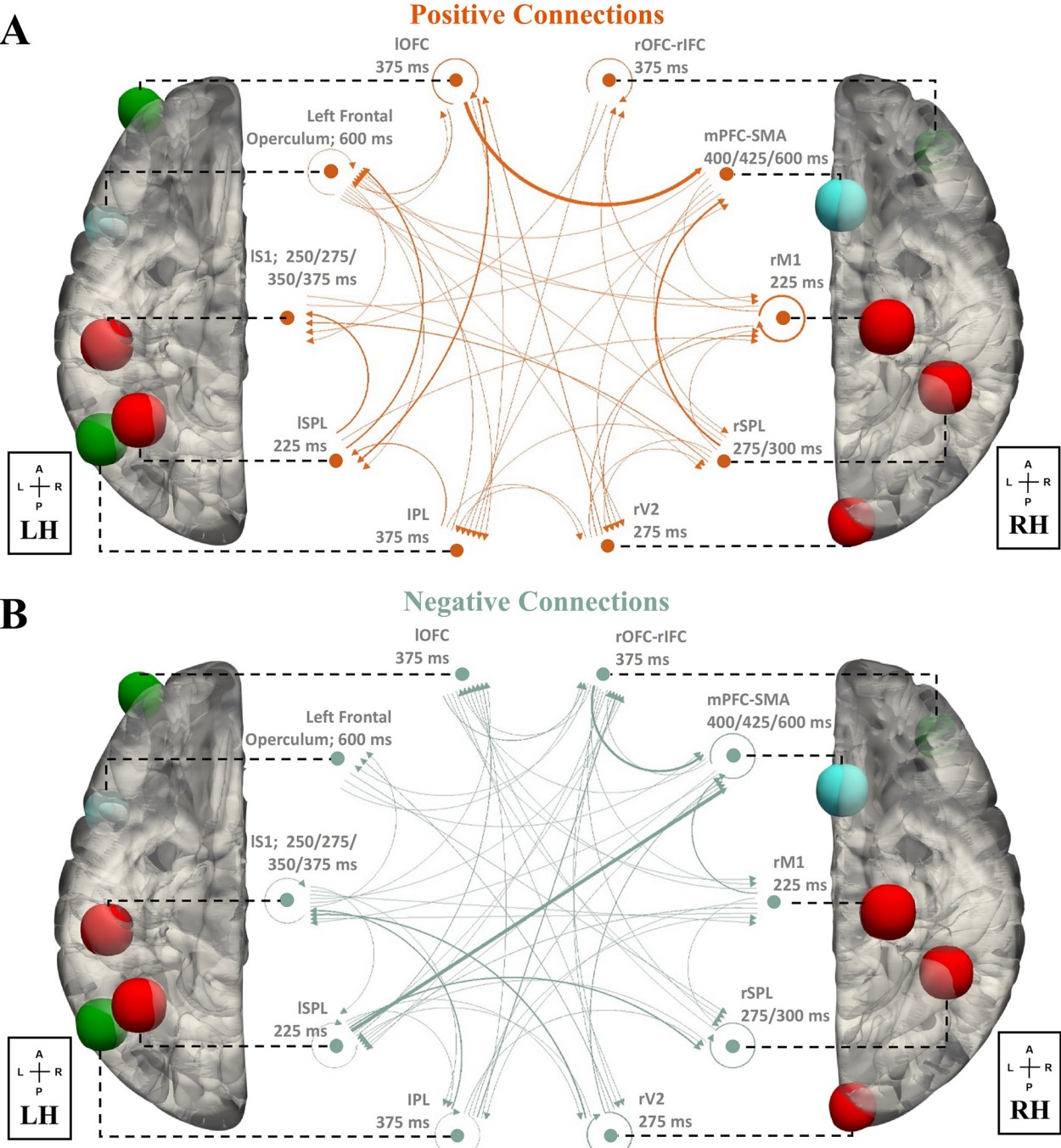

**Fig 4. Effective connectivity (EC) across salience processing nodes (Bayesian parameter averaging, $\alpha < 0.05$, Bonferroni corrected).** (A) positive EC, (B) negative EC. By leveraging the high temporal information in the EEG data, an effective connectivity state-space model was fit with the salience processing nodes. The arrow and thickness of the connecting lines correspond to the directionality and strength of EC, respectively. Dominant influence is observed in the connections of lSPL, lOFC, and mPFC-SMA. The results here reflect mean group effect. Node colors denote timing of involvement (early-time: red; middle-time: green; late-time: blue).

## Relationship between effective connectivity and pupillary response

Having demonstrated the directional interaction between the nodes modulated by salient stimuli, we next investigated the network-level EC, based on previously defined salience

processing networks. To characterize the network-level connectivity in the positive and negative connections, for each subject, we computed positive and negative network connection strength as the sum of all positive and negative connection parameters from one network node set to the other network node set (or to itself as self-connection network strength), respectively. Next, we assessed the relevance of network-level EC strength to TEPR, by computing the Pearson correlation between positive (or negative) network connectivity strength and TEPR at the between-subject level. We found a significant correlation between the late-to-early positive network connectivity strength and TEPR (r = 0.6352, p = 0.0035; Fig 5A; after controlling RT and model evidence lower bound (ELBO): r = 0.6347, p = 0.0035; ELBO was used as an evaluation of model fitting), whereas other network interactions did not show a significant correlation (Bonferroni correction; results in S1 Text). This significant relationship still holds at the between-run level, if the runs were pooled across all subjects (r = 0.4124, p = 0.0002). This outcome suggests that the TEPR is associated with the positive network couplings from the late-time network to the early-time network in the processing of salient stimuli, indicating a TEPR associated brain networks excitatory feedback (late-to-early) signal.

## Involvement of locus coeruleus in salience processing

Given substantial evidence that pupil diameter is tightly coupled to the neuronal activity in the LC [8, 11], pupil diameter has been used as an index of the LC activity [15]. With the observed correlation between late-to-early network feedback signal and TEPR (Fig 5A), we therefore hypothesized that the LC might play a role in the interactions between large-scale cortical networks. To test the involvement of the LC in salience processing, we first examined the functional connectivity between the LC and salience processing nodes (see S1 Text for details of the LC localization and BOLD signal extraction). The LC showed significant functional connectivity with lSPL, lS1 (lSPL: t = 2.64, p = 0.017; lS1: t = 3.80, p = 0.001; nodes of the early-time network and DAN) and mPFC-SMA (t = 3.15, p = 0.006; node of the late-time network and SN), however, there are no significant results between the LC and the other salience processing nodes (results in S1 Text). As a sanity check, whole-brain temporal signal-to-noise ratio analysis was performed with results in S7 Fig, and the seed-based whole-brain FC results of the LC are included in S8 Fig. With the functional couplings to the nodes of both early-time and late-time networks, this result indicates that the LC might be an important factor in the directional interactions between these two networks. This result is also consistent with the observation that lSPL and mPFC-SMA are the hubs in the modulated EC, rendering their importance in salience processing.

Following these results on the involvement of the LC with the late-to-early network feedback signal and the nodes of both DAN and SN (lSPL, lS1 and mPFC-SMA), and given the vast amount of literature on the triple-network model [3] of the DAN, SN, and default mode network (DMN), the final question we asked was whether this feedback signal reflects the network switching function of the SN, and whether pupil-indexed LC system is associated with the interactions between these three large-scale cortical networks. Thus, we fit the EEG data with the effective connectivity state-space model including the nodes of SN, DAN, and DMN, defined by the HCP-MMP (Human Connectome Project Multi-Modal Parcellation) atlas [25] (see S1 Text and S9 Fig for details of SN, DAN, and DMN nodes definition; EC results in S10 Fig). As expected, we found a significant positive correlation between the salient stimuli modulated SN-to-DAN positive network EC strength and TEPR (r = 0.6804, p = 0.0013; Fig 5B; after controlling RT and ELBO: r = 0.5949, p = 0.0072). This finding aligns with our previous results on the involvement of the LC in the late-to-early network feedback signal and the nodes of both DAN and SN. Interestingly, we also observed a significant negative correlation between the salient stimuli modulated SN-to-DMN negative network EC strength and

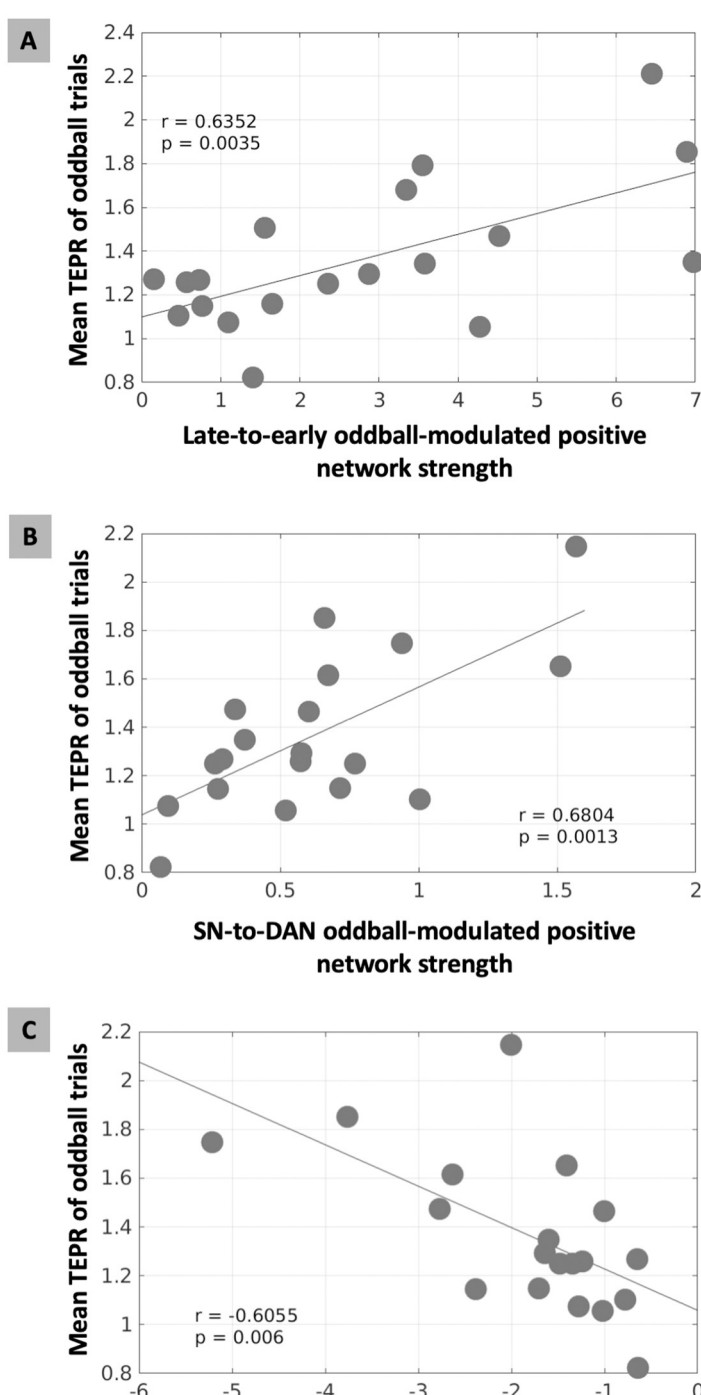

**Fig 5. Brain-pupil relationships of the cortical network-level effective connectivity (EC) and task-evoked pupillary response (TEPR) in salience processing.** (A) The oddball-modulated positive EC strength from the late-time to early-time network correlated with higher TEPR of oddball trials (p < 0.0035). In (B) and (C), to test the associations between pupil measurements and the triple-network model (SN, salience network; DAN, dorsal attention network; DMN, default mode network), we computed EC across nodes of these networks. (B) The oddball-modulated positive EC strength from SN to DAN correlated with higher TEPR of oddball trials (p < 0.0013). (C) The oddball-modulated negative EC strength from SN to DMN correlated with higher TEPR of oddball trials (p < 0.0060).

TEPR (r = -0.6055, p = 0.0060; Fig 5C; after controlling RT and ELBO: r = -0.6820, p = 0.0013). These significant relationships still hold at the between-run level, if the runs were pooled across all subjects (SN-to-DAN connection: r = 0.4051, p = 0.0002; SN-to-DMN connection: r = -0.2311, p = 0.0356). To eliminate the possibly alternative models that differ in the direction of information flow between SN, DAN, and DMN, we performed the same analysis between TEPR and all the other inter-network connections, and there is no significant relationship between them (results in S1 Text). These results are in line with the previous studies on the function of SN for the switching between anticorrelated networks [1, 26, 27]. In summary, the results indicate that the LC is involved in the switching between cortical networks.

## Discussion

### Spatiotemporal brain networks in salience processing

We used EEG-informed fMRI analysis to map the spatiotemporal dynamics of neural substrates in salience processing. Specifically, the STV temporal information in the EEG was extracted at different time windows spanning the trial to explain the variance in the fMRI signal. This approach has been widely used to study a broad range of cognitive functions and human behaviors [28, 29]. Compared to conventional fMRI analyses, EEG-informed fMRI analysis allows us to temporally dissociate the stimuli-evoked brain activation, or even identify regions absent in the conventional analyses (canceled out due to a temporal integration effect) [29, 30]. This work extends this approach by introducing the functional connectome into the framework for mapping the underlying spatiotemporal network organizations of these neural substrates. Functional connectome has been shown as a reliable approach in elucidating intrinsic brain organizations [31] and modeling cognitive task activation [24]. In this study, based on the STV EEG-informed fMRI analysis and functional connectome network localization, we observed a spatiotemporal intrinsic network organization of the neural substrates in salience processing (Fig 6). The involvement of these nodes and networks in an auditory oddball task is consistent with prior studies: DAN, motor network, ECN and SN [6], and visual network [30]. In this study, the FC analysis was used only to map network organizations of brain areas,

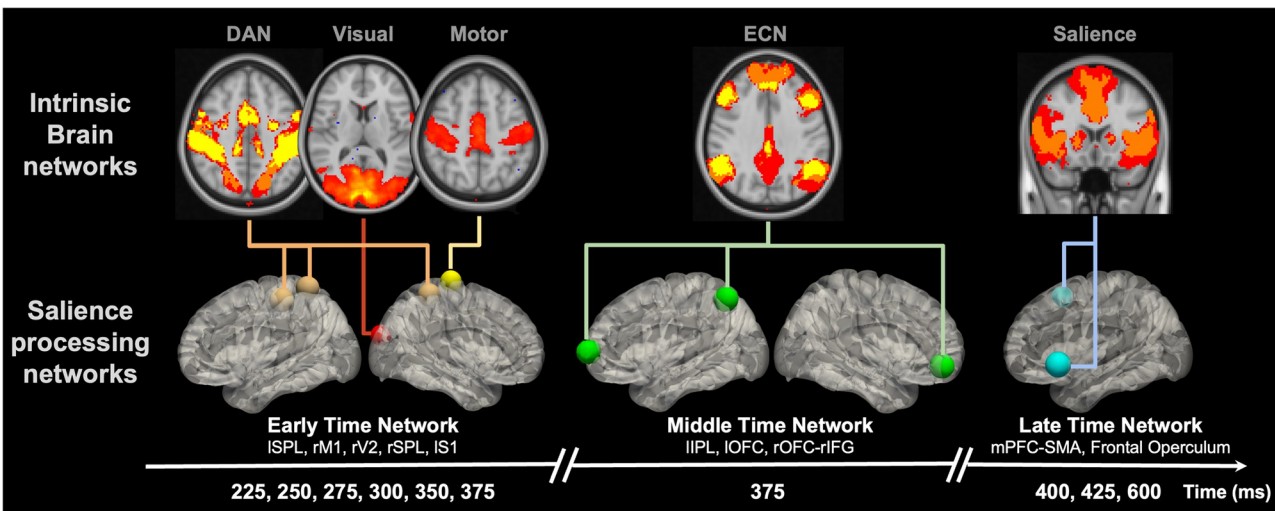

**Fig 6. Neural cascades of salience processing and the spatiotemporal network organizations of salience processing nodes.** Previous seed-based and node-by-node functional connectivity results suggest both spatial and temporal network organizations of the identified salience processing nodes, respectively. We hypothesized that the node-specific involvement of these functional networks might indicate a crucial role of these nodes in the temporally evolved processes of salience signal and the relationships between these networks. ECN, executive control network.

instead of making inference on the interactions, due to its correlation nature and the low temporal resolution of fMRI data. Thus, we performed EC analysis with EEG data. Comparison between the functional and effective connectivity with simultaneous EEG-fMRI is interesting, however, it is out of the scope of this paper.

Given these observations, we inferred that the brain flexibly recruited specific nodes of distinct networks at different time windows spanning the trial according to the demand of specific cognitive processes and behavior responses. In the early-time windows, we observed a sustained activation of DAN subsystem from 225 ms to 375 ms, along with the rM1 at 225 ms and rV2 at 275 ms. Literatures have shown that DAN is associated with goal-driven attention and the role of linking them to appropriate motor responses [16]. The observed activation of the early-time network might reflect the functions of these systems in salience processing. For example, the coactivation of rM1 and lSPL at 225 ms might reflect the relationship between DAN and the motor network, and their role in linking stimuli and responses. Based on the previous evidence that DAN exhibits top-down influences on the sensory cortex [16], we hypothesized that the involvement of the visual network areas might indicate the modulation of attentional resources distribution [32]. This hypothesis was also supported by the observed self-inhibition EC in rV2 (Fig 4B). Similar to the reported temporal components underlying a visual spatial attention task in a Magnetoencephalography (MEG) study [33], we also observed the involvement of ECN nodes (375 ms) right after the early activation of parietal and visual areas. This network's node-specific involvement might indicate a crucial role of these nodes in the network and salience processing. For example, prior studies suggest the existence of common nodes (left orbital frontoinsula, mPFC, and right dorsolateral prefrontal cortex) between the ECN and SN [7]. Thus, the observed involvement of the left frontal operculum and mPFC-SMA in the late time might reflect their roles in the relationship between ECN and SN, which might facilitate the temporal transition from ECN to SN in the late time of the trial. Future studies are needed to investigate the specific functions of these nodes in the relationship between brain networks.

## Cognitive control in salience processing

Based on the anatomical locations of the nodes in the network, ECN and SN have also been named lateral frontoparietal network (L-FPN) and midcingulo-insular network (M-CIN), respectively [34]. Previous studies have shown that the ECN/L-FPN and SN/M-CIN are two prominent cognitive control networks, supporting the goal-directed cognition and behavior [35, 36]. In these studies, L-FPN and M-CIN were named as frontoparietal network and cingulo-opercular network. To keep the terminology of the brain networks consistent in this study, we referred to ECN/L-FPN and SN/M-CIN following the guidelines in [34]. SN (cognitive domain name) and M-CIN (anatomical name) contains these core regions: bilateral anterior insula and anterior midcingulate cortex. ECN (cognitive domain name) and L-FPN (anatomical name) contains these core regions: lateral prefrontal cortex, anterior inferior parietal lobule, and intraparietal sulcus. ECN/L-FPN and SN/M-CIN are coactivated together as the task-activation ensemble [7] in goal-directed behaviors and tasks. However, converging evidence from the literature indicates distinct roles of ECN/L-FPN and SN/M-CIN in goal-directed behaviors. ECN/L-FPN acts as a flexible coordinator of goal-relevant information [37], and may underlie phasic control such as initiating exogenously triggered control, adaptive adjustments, and executive functions [38, 39]. Whereas, SN/M-CIN is related to stable maintenance of task-set [36] and tonic alertness [40], and has also been proposed a role in lending processing resources to help other goal-relevant networks [37].

In the present study, by leveraging STV temporal information in the EEG to tease apart the temporal neural processes in the goal-directed salience processing, we dissociated the stimuli-

evoked coactivation of ECN/L-FPN and SN/M-CIN into two distinct temporal components (subsystem of ECN as the middle-time network deactivated at 375 ms, subsystem of SN as the late-time network deactivated during 400–600 ms). Along with this temporal dissociation, ECN/L-FPN and SN/M-CIN seemed to act with distinct roles in the salience processing, where ECN/L-FPN was deactivated preceding the behavior response (group-averaged RT: 404 ms) and the deactivation of SN/M-CIN. And ECN/L-FPN acted as a phasic control (deactivated only at 375 ms), which might provide the rapid control initiation to SN/M-CIN (deactivated starting 400 ms). This is also supported by the EC results that the oddball-modulated EC (S5 Fig) showed a stronger connectivity strength from the subsystem of ECN/L-FPN to SN/M-CIN (mid-to-late; positive: 0.2523; negative: 0.3145) compared to the connection from the subsystem of SN/M-CIN to ECN/L-FPN (late-to-mid; positive: 0.0165; negative: 0.1126). These results align with the interactive dual-networks model of ECN/L-FPN and SN/M-CIN in [38]. This rapid control initiation might be mostly driven by the EC from lOFC to mPFC-SMA, as indicated by the strong connection strength (Fig 4A). Whereas, SN/M-CIN was deactivated at multiple time points (400, 425, 600 ms) right after the response, indicating a stable maintenance of tonic alertness, which might facilitate the better detection performance in the next coming trial [40]. The involvement of SN/M-CIN regions in processing salient stimuli was reported by [4, 5], where the anterior cingulate and the supplementary motor areas had been proposed a role in anticipation of forthcoming stimuli. We propose that the involvement of SN in the late time windows might allow the brain to disengage the current trial and maintain the preparedness for the next upcoming stimulus. As proposed in a single-unit recording study, pre-SMA (mPFC-SMA) is associated with task switching by first suppressing irrelevant task-set and then boosting a controlled response with the relevant task-set [41, 42]. In the present study, we found mPFC-SMA deactivated starting at 400 ms, which might reflect a suppression or inhibition of the current trial encoded task-set. With the proposed role of pre-SMA (mPFC-SMA) in conflict monitoring [41, 43], in the changing environment after the response, its involvement allows the brain to resolve the conflict between the current trial encoded task-set and the new environment, which facilitates task disengagement. This task-set suppressing and boosting role of the SN is consistent with the network switching theory [27]. It is worth noting that, in the traditional fMRI analysis of oddball effects (S4 Fig), regions in the SN were identified as significantly activated clusters, whereas, our single-trial variability EEG-informed fMRI analysis results suggest that the SN was deactivated in response to oddball events in the late window (Figs 1 and 6). These observations indicate that though the mean response of the SN is stronger for the oddball trials compared to the standard trials, our EEG-informed time-resolved analysis shows that late in the trial, when an oddball is less discriminating from a standard, as measured by EEG, the SN response is stronger to the oddball trial. We believe this result supports the proposed "windshield wiper" role of the SN in the literature [40], and a stronger response in the SN might suppress the discriminating processes of the current trial, facilitating task disengagement. However, this hypothesis warrants further investigation. In the anticipation of upcoming inputs, SN may employ such a mechanism to increase preparedness by clearing currently ongoing activity in multiple cortical areas. The present findings provided more evidence for the functions, relationship, and timescales of these two cognitive control networks (i.e. ECN/L-FPN and SN/M-CIN).

### Linking networks effective connectivity and pupillary response: LC is associated with network reset

The LC-NE system has been proposed to modulate neural gain, attention, and arousal [8]. Pupil diameter fluctuations, as a proxy of LC activity [11, 12], have been used to investigate

how the ascending neuromodulator from the LC-NE system influences the cortex [10]. There is growing research on the relationship between brain measurements and pupil diameter, with evidence suggesting that pupil diameter fluctuations are associated with cortical membrane potential activity [44], EEG P300 component of the ERP [10], fMRI BOLD signal in the DMN, SN, thalamus, frontoparietal, visual, and sensorimotor regions [45, 46], overall functional connectivity strength during exploration [47], and global fluctuations in network structure [48]. These findings shed light on the understanding of brain-pupil relationships and cortico-subcortical associations, however, the role of the LC-NE system in the connectivity and interaction between specific brain networks within the context of a goal-driven task (e.g. salience processing) is less well understood. Here, by leveraging the high temporal resolution of the EEG, we used a state-space model for inferring the EC between brain networks involved in salience processing. We observed a very strong relationship between the late-to-early positive network interaction strength and TEPR (Fig 5A). In the oddball paradigm with motor response, studies have shown that TEPR reflects not only bottom-up mechanisms but also top-down higher-order processing [6, 13]. Given the directionality of this TEPR-related network interaction, we propose that the phasic LC activity (indexed by TEPR) is associated with a feedback (top-down) signal from the late-time network (nodes of SN) to the early-time network (nodes of DAN, visual, and sensory motor network). Besides the close relationship between the LC activity and pupil diameter fluctuations, SN areas also exhibited close links to the LC-NE system and pupil measurements. The LC-NE system has shown to receive projections from the anterior cingulate cortex (ACC) and anterior insula (AI) [8, 16], and has robust functional connectivity with these SN nodes [17, 18]. In an intracranial EEG study [49], both the spontaneous and task-evoked activations in the anterior insula are linked to the dynamics of pupillary dilation. The SN and pupil measurements have both been associated with task demands, efforts, difficulty [50, 51], uncertainty and surprise [13], conflict and error processing [52], and anxiety [7, 53]. The involvement of SN, as a subsystem of the ventral attention network (VAN) [34], aligns with evidence showing the involvement of VAN in both bottom-up stimulus saliency and top-down internal goals [16, 54].

Even with widespread projections of LC neurons throughout the cortex, recent studies suggested that there are substantial specificity and heterogeneity in the projections [20, 47, 55]. For example, regions in DAN receive dense LC-NE inputs [56]. Our functional connectivity analysis suggests that the LC-NE system, with connections to both SN and DAN nodes, might play an important role in the top-down control from SN to DAN along with other early-time network nodes. Our results strongly support the network-reset theory, which proposes that the VAN (SN as a subsystem) marks behavior transitions and facilitates a network reset signal along with the phasic LC-NE activity (indexed by TEPR), to reconfigure the DAN (part of the early-time network) for settling the brain into another state in the new environment situation [14, 16]. This theory also aligns with the previous discussions on the synchronized timing of the behavior response/transitions and the deactivation of SN areas, and the proposed preparedness role of the late involvement of SN areas. In light of recent studies in the LC-NE system effects on brain network reconfiguration [21], more studies are needed to investigate the role of LC-NE system in the interaction between brain networks [20, 57].

## SN, DAN, and DMN in salience processing: LC plays a role in network switching

The anticorrelation between the DAN and the DMN has been characterized as a vital aspect of the human brain functional organization and dynamics [31], with DAN and DMN controlling environmentally directed and internally directed cognitive processes, respectively [16, 31].

Converging evidence suggests that the nodes of the SN are at the apex of the cortical hierarchy between these two anticorrelated networks [26], with a critical role in the dynamic switching between them [58]. A triple network model has been proposed for these three core neurocognitive networks [3], serving as a networks framework for understanding psychopathology [3] and cognitive aging [59]. However, the neural mechanisms underlying dynamic switching, and how can the SN have such a wide spread access to DAN and DMN for coordinating the switching between them, are not well understood. Here, along with the close relationship between the SN and the LC-NE system discussed in the previous section, it is plausible to hypothesize that brainstem nuclei, such as the LC, may play a role in the dynamic switching of large-scale brain networks through the release of neuromodulatory neurotransmitters. Neuromodulation models of the LC have been proposed. In the adaptive-gain theory [8, 15], the LC receives task utility information from the ACC (SN node) and OFC, producing NE release at cortical target sites and adjusts the gain. The network glutamate amplification of noradrenaline (GANE) model [57, 60] proposed that the SN recruits LC firing to enable NE local concentration modulation, accompanied by in parallel enhancement and suppression of large-scale brain networks. In support of our hypothesis, i.e. the LC is associated with the network switching function of the SN, we found that increased TEPR (index of phasic LC activity) is associated with a stronger positive EC from the SN to the DAN, and a stronger negative EC from the SN to the DMN (Fig 5). This result confirmed the previous findings that the SN initiates the dynamic switching, and to our knowledge, this is the first study to show the integrative role of the LC-NE system and the SN in the dynamic switching between anticorrelated networks. This suggests a cortico-subcortical integrated network reorganization (CS-INR) system, involving both the SN and the LC-NE system in the network reorganization (reset and dynamic switching) between the DAN and DMN (Fig 7).

Regarding the temporal profiles of the task responses in SN, DAN, and DMN, our results align with several intracranial EEG studies [61, 62], where they found the responses reached the fastest, intermediate, and slowest speed in the DAN, SN, and DMN, respectively. In our study, we also found an earlier involvement of the DAN (starting at 225 ms), preceding the responses in the SN starting at 400 ms. Interestingly, in our previous simultaneous EEG-fMRI study [63], we identified significant responses in the DMN at a relatively late time window (525 ms), supporting the temporal order of responses in DAN, SN, and DMN as reported in these intracranial EEG studies. However, these findings in the task-evoked responses do not imply interactions among these networks. In this study, our effective connectivity analyses extend such frameworks of networks' temporal dynamics, by proposing that the network switching signal from the SN to the DAN might reflect a late-to-early feedback signal. Here, the effective connectivity analysis results reflect the interactions between the networks regardless of the time windows. In support of the claims made previously, the strong relationship between the SN-to-DAN EC and TEPR suggests an important role of the LC-NE system in this network reset (or circuit-breaker) top-down control signal. Based on the involvement of the SN nodes as the late-time salience processing network (Fig 6), and its higher hierarchy among the DAN and DMN [26], the CS-INR is thought to be involved in the late phases of the salience processing, reflecting a mark of behavior transition, task disengagement, and preparedness, initiated by the deactivation of SN. However, our findings do not rule out the possibility that the CS-INR system might also be involved at the other phases of the trial. For example, a recent study has found that the DAN, SN, visual and frontoparietal regions are involved in the early phase of the re-orienting, which has been interpreted as a network reset signal [64]. Different from our findings, these results might reflect the reorienting modulation in task engagement or bottom-up stimuli processing. Consistent with our hypothesis, the LC activation has been shown to be more closely aligned with the behavioral response than the stimulus onset [8, 57],

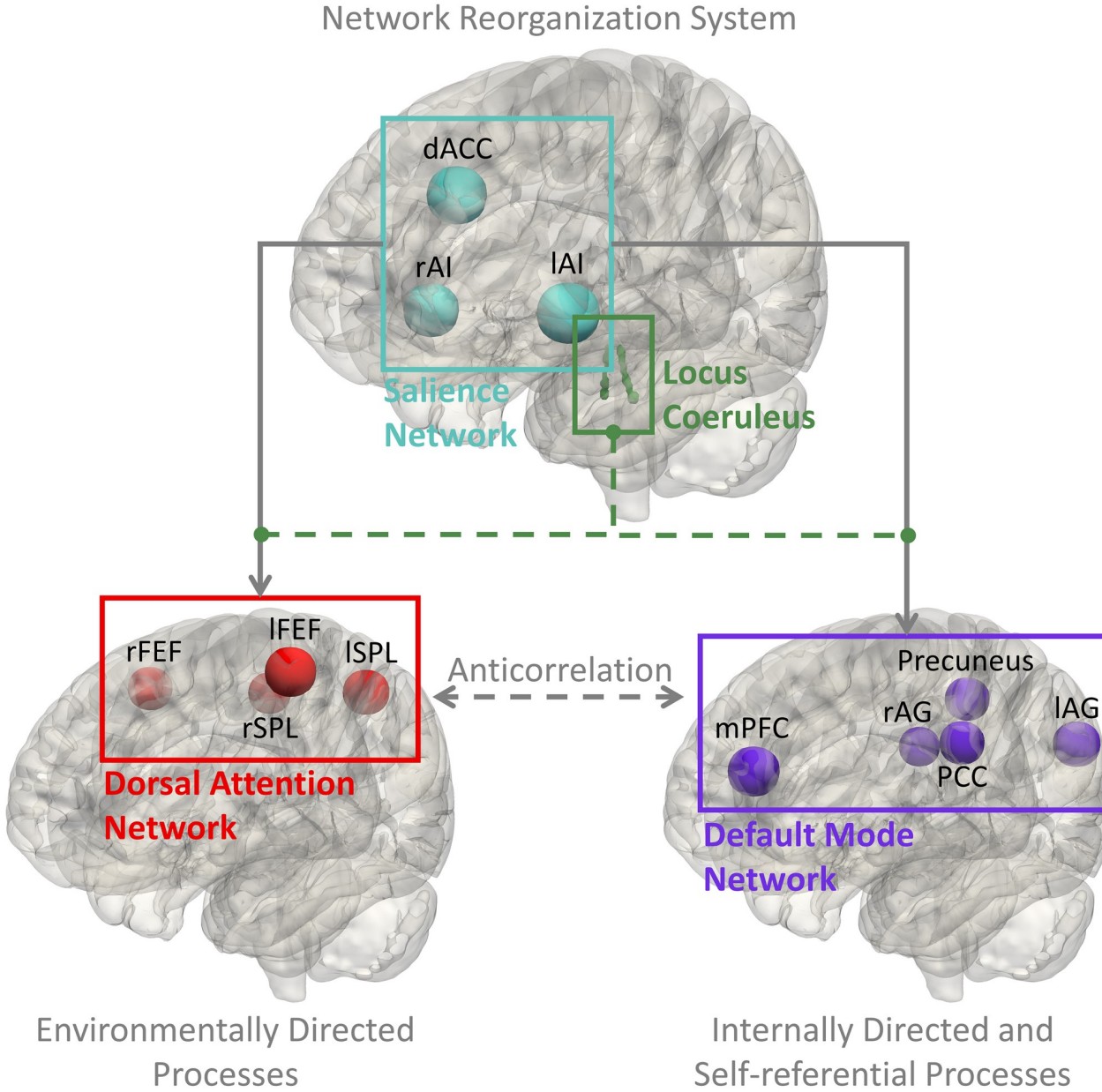

**Fig 7. Cortico-subcortical integrated network reorganization (CS-INR) system.** Previous brain-pupil relationships results aligned with the network switching model of SN in the literature, and also showed the role of the locus coeruleus norepinephrine (LC-NE) system in the network reset and the dynamic switching between anticorrelated networks (SN-to-DAN and SN-to-DMN). In support with the literature [14], we hypothesized that the reset and switching might be modulated by the release of the NE, as an effect of the ascending neuromodulation, which indicates that the SN and LC-NE system might cooperate and share an integrated role in salience processing. dACC, dorsal anterior cingulate cortex; lAI and rAI, left and right anterior insula; lSPL and rSPL, left and right superior parietal lobule; lFEF and rFEF, left and right frontal eye fields; mPFC, medial prefrontal cortex; PCC, posterior cingulate cortex; lAG and rAG, left and right angular gyrus.

and our previous work has also demonstrated that the DMN is involved in the late phase of the trial in a target detection task [63]. The DMN has been implicated in future planning [65], task switching [66], attention shifts [67], as well as other functions [65], and the LC-NE system has been proposed to modulate the DMN as a neural modulator of mind wandering [68]. Our results are consistent with hypothesis that the LC-NE system modulates the connectivity

between the DAN and DMN [69], and also the proposed role as a 'master switch' [70]. Potential candidate mechanisms of the LC-NE modulation on the DAN and DMN are: 1) The LC-NE system modulates them through the heterogeneous spatial distribution of NE receptor types and densities across the cortex [19, 71]; 2) The heterogeneity in LC cell populations might be responsible for the targeted modulations of specific cortex areas [55, 72], for example, a modular organization in the LC with distinct efferent neural projection patterns has been reported [57, 73]; 3) The LC-NE system might interact with other subcortical nuclei in the modulation [20], such as thalamus, which has been associated with the LC activity [18], SN activity [7], and the modulation of cortical networks connectivity [74, 75]. The work may inform future studies of the neural mechanisms underlying the LC-NE modulation on the cortex.

The close relationship between the LC-NE system and the SN has been investigated thoroughly in the literature, for example Herman et al. found that the noradrenergic activation during acute stress results in the changes in the functional connectivity strength within the SN, and these changes were inferred as network reconfiguration in their study [17]. Their findings established a causal link from the LC-NE system activity to the activity of the SN, however the interaction between the SN and other brain networks, and how the two systems cooperate in an integrative framework are still unclear. Whereas, our study suggests that the LC-NE system might be responsible for the network-level interactions between the SN and other brain networks. The CS-INR network model proposed here bridges the gap between the network reset model of the LC-NE system and the network switching model of the SN, which could potentially serve as a cortico-subcortical network reorganization paradigm for understanding the neural dynamics underlying various cognitive functions, such as salience processing. With the neuronal basis of fast control in the von Economo neurons (VENs) of SN [27] and the spatially diffuse projections from the LC [8], the CS-INR system is ideally suited to a variety of complex cognitive processes. Although our results shed light on the relationship between the LC-NE system and the cortical network reorganization in the context of an auditory oddball task, we speculate that the CS-INR system might play a more general role in cognitive functions, such as adaptations in environmental volatility [53, 76], brain state switches/variations [44, 77], cognitive control [15, 37, 48, 78]. Further work is needed to uncover the implications of the CS-INR model in various cognitive processes and neurological diseases. For example, previous studies have associated the activity in the DAN and DMN regions with exploitation and exploration, respectively [79]. And it has been proposed that the SN and the LC-NE system may play a role in the switching from exploitation to exploration [8, 79], hence, it will be interesting to test the CS-INR model in the exploration and exploitation tasks.

## Broader implications of a simultaneous pupillometry-EEG-fMRI study

In this study, we deployed a framework involving simultaneous recordings of pupillometry, EEG, and fMRI to investigate neural processes and interactions in salience processing. The high spatial resolution of fMRI data and functional connectivity analysis were utilized to map the neural substrates and the functional organizations. The EEG data with high temporal resolution, the single-trial analysis, and the effective connectivity state-space model were used to temporally 'tag' the neural substrates and infer the directional interactions. As a proxy of the LC activity, pupillometry was included to study the cortico-subcortical associations. The multimodal methodological approaches and the CS-INR network model proposed here might promote further investigations on the brain dynamics underlying various cognitive processes and neurological diseases. For example, the role of the LC in cognitive control is intriguing but has not been fully explored. This might be due to the gap in current knowledge between the

network models of the cortex and the models of the LC, such as how the LC's function in network reset is related to the network switching function of the SN. Besides their critical contribution to attentional processing as demonstrated in this study, the CS-INR network model proposed here, or other cortico-subcortical network models, is also critical in the understanding of neurological diseases, such as Alzheimer's disease (AD). As the first brain region in which AD-related pathology appears, the LC has been associated with cognitive decline and aging [80]. The studies of the LC and pupil diameter fluctuations, and their interactions with the cortical networks have important implications for the understanding of neurological diseases, such as AD. For example, a recent study showed that the LC in the older population has reduced interaction with the SN, suggesting subsequent impairment in the initiation of network switching, and inferior ability in prioritizing the importance of incoming events [81].

In addition to the other work from our group where we used this multi-modal approach to examine the relationship between LC activity, pupillary response, and cortical dynamics [82], there are two other very recent instances where this simultaneous triple-modality data acquisition was reported [83, 84]. Together with the work presented here, all four studies showcase unique analyses and insights that could be harnessed from simultaneous pupillometry-EEG-fMRI. For instance, while this work and Hong et al. used an asymmetric fusion approach, Groot et al. applied a symmetric fusion to the multi-modal data through a support vector machine, in order to investigate neural signatures of task-unrelated thoughts. Taken together, we believe that despite the technical challenges, this simultaneous multi-modal approach holds great value and potential in unraveling cortical dynamics at various levels.

## Limitations

In this study, we used effective connectivity to infer the directional couplings between brain networks. In the effective connectivity state-space model, the temporal dependence among the dynamics of latent neural states of different brain areas was modeled as a multivariate autoregressive process. For example, the effective connectivity estimates how the future neural states in one brain area are influenced by the current neural states in another brain area, and how the external experimental perturbation can modulate these couplings. Thus, the directionality of information flow was inferred based on temporal forecasting and control theory, and the 'directional interaction' in this paper is limited under the assumptions of our effective connectivity state space model. Future studies using simultaneous pupillometry, neuroimaging, and transcranial magnetic stimulation would be interesting to explore the relationship between pupil diameter fluctuations and brain networks couplings, and potentially to provide stronger evidence on the 'directionality' of brain networks interaction. Furthermore, the directionality in the cortico-subcortical interactions between the LC and cortex regions remain unclear. Based on the involvement of the SN in the late time of the trial observed in our data, we hypothesized that the network switching might be modulated by the release of the NE, as an effect of the ascending neuromodulation. Our hypothesis aligns with the findings in [17], which provided evidence on the causal link from the LC-NE system activity to the activity of the SN. Whereas, in the adaptive gain theory, the LC is proposed to receive inputs from ACC and OFC, with the release of NE at cortical target sites [8]. Further investigations will be needed to better understand the directionality in the interactions between the LC and cortex regions.

Pupillometry has long been used to index the LC activity in previous studies [8, 10, 13, 15] and also in the current study, though it is challenging, as other neural circuits are involved in controlling pupil diameter as well [13]. For example, shifts of attention are mediated in part by the superior colliculus (SC). In the previous studies, the LC rather than the SC showed

neural spiking responses to unexpected auditory events [11]. In an oddball task fMRI study, the pupil size fluctuations have been associated with the BOLD activity in the LC [12]. In this study, we utilized this well-studied oddball paradigm, where the coupling between pupillometry and the LC activity has been shown, to investigate the pupil-indexed activity in the LC. However, this does not rule out the possibility that other subcortical nuclei, such as thalamus, or other neuromodulators, such as acetylcholine, might contribute to pupil diameter fluctuations or interact with the LC-NE system. Further investigations are needed for direct neuroimaging of the LC, however, it is challenging due to the excessive physiological noise and distortion in brainstem imaging [85], and the difficulty in the localization of the LC [18]. Additionally, the present study did not examine the relationship between the LC activity and pupillary response, as well as its relationship to the EEG measurement of cortex processes. Another study from our group highlighted these additional analyses and comprehensive examinations of LC activity and their associations [82, 86]. In this study, we examined the relationship between effective connectivity and pupil measurement at the inter-individual level as shown in Fig 5. Future studies are needed to replicate our results at the intra-individual level. However, care should be taken when fitting the effective connectivity model without enough data points. And more experimental sessions and data acquisition for each individual may be needed.

In this study, we made inferences on the role of the pupil-indexed LC activity in salience processing, based on their interaction with the task-related neural substrates. However, besides the attentional processing of salient stimuli, pupil-indexed LC activity has also been associated to changes in arousal. Though these two LC associated processes, i.e. attention and arousal, have been shown to be independent [9], an important future direction will be accounting for the LC associated arousal, and assessing its relationship to the cortex. In future studies, novel tasks can be devised to dissociate pupil-indexed LC activity in attention and arousal.

## Conclusions

In summary, the present multimodal study analysis of pupillometry-EEG-fMRI reveals a coordinated neural cascade during salience processing, involving brain regions in the dorsal attention, visual, motor, executive control, and salience networks. Additionally, the study identified the network organization and effective connectivity between these brain regions, which were found to be associated with salient stimuli evoked pupillary response. These findings are in line with the proposed function of the pupil-indexed LC-NE system in network reset [14, 16]. To advance understanding of the interactions between neuromodulatory systems and intrinsic brain networks in relation to cognition, the study presented additional analyses on the associations between the LC-NE system and networks including the SN, DAN, and DMN, where the results provide the first evidence in humans for the relevance of the LC-NE system to the function of the SN in the dynamic switching between anticorrelated cortical networks. This work has important implications for multimodal neuroimaging data analyses, brain-pupil relationships, attentional processing, cognitive control networks, and network models of neurological diseases.

## Materials and methods

### Ethics statement

The experimental design of our study and the recruitment process were approved by Columbia University institutional review board. All participants have provided informed consent to participate in the study, and written consent was obtained from the participants.

## Overview

In this section, we describe the analyses of the multimodal pupillometry-EEG-fMRI data, specifically focused on experimental design, data acquisition and preprocessing, EEG single-trial analysis, EEG-informed fMRI analysis, fMRI functional connectivity analysis, EEG effective connectivity analysis, and brain-pupil correlation analysis. A flow chart is included in S11 Fig to illustrate the steps of data processing and single-modality/cross-modality data analyses.

## Participants and experimental design

Twenty-five healthy young subjects were recruited in this study and six of them were excluded from further analyses due to 1) missing neuroimaging data; 2) abnormality in the acquired neuroimaging data; 3) excessive movement; 4) inability to complete the task. Exclusion criteria were pre-established. Data from the remaining nineteen subjects (mean age ± SD = 25.9 ± 3.6 years, female/male = 13/6) were included in the analyses. All subjects had normal or corrected-to-normal vision and no history of psychiatric illness or head injury. We used a convenience sampling procedure through recruiting volunteer subjects from Columbia University and nearby areas. The sample size was based on previously published simultaneous EEG-fMRI studies using a visual oddball task with seventeen subjects [63] and a decision making task with twenty-one subjects [87].

An auditory oddball paradigm with 80% standard and 20% oddball (target) stimuli was performed, where standard stimuli were pure tones with a frequency of 350 Hz, and the oddball stimuli were broadband (laser gun) sounds. We chose an auditory (instead of visual) oddball paradigm to avoid the effects of luminance changes on the measurements of task-evoked pupillary response. We randomized the presentation of oddball and standard trials and trial order. The inter-trial intervals were in the range between 2 s and 3 s drawn from a uniform distribution, and each stimulus lasted for 200 ms. Subjects were first trained outside of the scanner to learn and perform the task comfortably and accurately on short training runs. All subjects performed the task correctly during training. During the data acquisition, stimuli were presented through MR compatible earphones, and subjects were instructed to maintain the fixation on the screen to a fixation target, and press a button (MR-compatible button box; PYKA, Current Designs, PA, USA) with their right index finger as soon as they heard the oddball sound. And subjects were instructed to ignore standard tones. Every subject was scheduled to complete five runs (105 trials per run), with an average of 4.7 runs per subject (range from three to five, SD = 0.7 runs) acquired in the experiment. The auditory oddball experimental task paradigm is illustrated in the first row of S2 Fig, where the first five trials were constrained to be standard stimuli, and no consecutive oddball trials was allowed.

## Data acquisition and preprocessing

A 3T Siemens Prisma scanner was used to acquire pupillometry, EEG and fMRI with a 64-channel head coil. Pupillometry was recorded with a MR-compatible EyeLink 1000 Plus in Long Range Mount, at a sampling rate of 1 kHz. EEG was recorded with a 64 channel BrainAmp MR Plus system (Brain Products, Germany), at a sampling rate of 5 kHz. The 64 channels include 63 cap electrodes and 1 ECG electrode in an extended 10–20 configuration with ground electrode at AFz and reference electrode at FCz. Functional MRI data were collected with T2*-weighted echo planar imaging (EPI) interleaved slice acquisition (TR = 2100 ms; TE = 25 ms; voxel size $3 \times 3 \times 3$ mm; Matrix Size = $64 \times 64 \times 42$; 150 volumes). For localization and registration purposes, we collected T1-weighted structural image (MPRAGE, TR = 2300 ms; TE = 3.95 ms; voxel size = $1 \times 1 \times 1$ mm; Matrix Size = $176 \times 248 \times 256$) and T2*-weighted high-resolution EPI (TR = 6000 ms; TE = 30 ms; voxel size $2 \times 2 \times 3$ mm; Matrix

Size = 96 × 96 × 42; single-volume). For the localization of the LC, we also collected neurome-lanin-sensitive MRI data using T1-weighted turbo-spin-echo (TSE) acquisition (TR = 600 ms; TE = 14 ms; voxel size 0.43 × 0.43 × 6 mm; Matrix Size = 416 × 512 × 5). Descriptions of pupil-lometry, EEG and fMRI data preprocessing are included in S1 Text. Data acquisition specifica-tions (along with the rigorous measures taken to ensure data quality) were described in detail in [82]. Specifically, proper synchronization between EEG recording and MR imaging was ensured via Brain Products' SyncBox. The SyncBox receives pulses coming from the scanner's gradient clock board directly, and can therefore synchronize the sampling rate of the amplifier with the scanner clock system [88]. And to ensure subject safety during simultaneous EEG and fMRI acquisition, the scalp and ECG electrodes were embedded with series resistors of 10 kOhm and 20 kOhm, respectively. During the experiment, electrodes' impedances were kept under 25 kOhm (including the built-in resistors on each electrode) to minimize the noise in EEG acquisition.

## EEG single-trial analysis

A single-trial analysis with the sliding window approach was performed on the preprocessed EEG signal amplitude [89]. Specifically, with a linear classifier maximally discriminating the target versus standard trials, a hyper-plane was learnt to project the multidimensional EEG signal into low-dimensional EEG single-trial variability discriminating components. Given the EEG signal, $y_i(t)$ at time $t$, where $i = 1, 2, T$ denotes trial index, logistic regression was used to learn the projection weights $w(\tau)$. The low-dimensional EEG STV discriminating components will be: $d_i(\tau) = \frac{1}{N} \sum_{t=\tau-\frac{N}{2}}^{\tau+\frac{N}{2}} w(\tau)^T y_i(t)$, where $N = 50ms$ denotes window width, the window center $\tau$ was shifted from 0 to 1000 ms with respect to the stimulus onset in 25 ms increments. For each temporal window, the classifier performance was assessed with the AUC using leave-one-out (LOO) cross-validation. A permutation test was used to obtain the significance threshold for the AUC (100 times of permutations for each subject), where trial labels were randomly permuted and LOO was carried out. The null distribution of AUC values was generated and a threshold of p < 0.01 was used. Details of the STV analysis are illustrated in the S2 Fig.

## STV EEG-informed fMRI analysis

We fit a GLM for each EEG STV time window. First-level GLM was performed independently on each voxel using multiple regression with five variables of interest. The regressors included: 1) event-related regressors with unmodulated height, and both onset and duration matched to the presence of the stimulus (one each for targets and standards); 2) RT variability regressor with unmodulated height, onset matched to the stimulus onset, and duration matched to the RT of the trial (orthogonalized with respect to the targets event-related regressor); 3) EEG STV regressors with height parametrically modulated using the demeaned EEG STV discriminating component, onset set to the time of interest $\tau$, and duration fixed to 100 ms (one each for tar-gets and standards, orthogonalized with respect to the corresponding event-related regressor, oddball EEG STV regressor was also orthogonalized to RT regressor); 4) confounds (motion parameters, temporal derivatives of the variables of interest). At each time window $\tau$, the demeaned output of the logistic regression classifier, as the EEG STV discriminating compo-nent $\tilde{d}_i(\tau) = d_i(\tau) - d_{mean}(\tau)$ was used to modulate the height of the EEG STV regressor box-car function. All regressors were convolved with canonical Double-Gamma hemodynamic response function (HRF). The preprocessed fMRI data were spatially smoothed with a Gauss-ian kernel of FWHM 5 mm, then were fit with the GLM resulting to five different statistical parametric maps which will be warped onto a standard space (i.e. MNI152) to be able to

perform group-level statistical analysis. For group-level statistical inference, we used FMRIB's Local Analysis of Mixed Effects (FLAME) from the FSL software package, where a mixed effect model was carried out, and the group-level statistical parametric maps were thresholded ($|z| > 2.3$, corrected cluster significance threshold of p = 0.05, Gaussian random field method). As the oddball EEG STV regressor was the primary regressor of interest (i.e., the regressor indicative of task-relevant processing), we only used cortical regions whose BOLD signal covaries with this particular regressor in the subsequent analyses.

## fMRI functional connectivity analysis

To control for physiological and motion-related noise, we regressed out motion-related nuisance signals from the preprocessed fMRI data (motion parameters included six standard head motion parameters, their temporal derivatives, and the squares of the above twelve motion parameters), as well as the signals in the left and right hemisphere white matter and lateral ventricles. To circumvent the error in spatial normalization and smoothing, we extracted the BOLD time series of the regions of interest (ROIs) in each subject's native functional EPI space. The ROI masks were transformed and warped into the subjects' EPI functional space with the estimated registration parameters. For each ROI, only the gray matter BOLD signal was extracted by computing the intersection between the ROI masks and the gray matter mask. And we extracted a single BOLD time series from each ROI by averaging the time series of all the voxels within the intersected mask. No spatial smoothing was carried out. Before computing the FC, to remove the effects of the task, we regressed out task-evoked activations (both standard and target) from the ROIs BOLD signals. In the seed-based FC analysis for network localization, we used a mixed effects model. The first-level seed-based FC was calculated based on the Pearson correlation between the time series of each ROI and the time series of each voxel in the brain. Then, each subject's FC map was transformed into z-score with Fisher's Z transformation. And the FC z-score map was thresholded at p < 0.01. In the group-level, one sample student's t-test was performed to obtain the significant seed-based FC map of each ROI (p < 0.001 uncorrected). Similarly, in the node-by-node FC analysis, the first-level FC was computed based on the Pearson correlation across the time series of ROIs. Then, the FC matrix was transformed into z-score, and carried out to the group-level one sample student's t-test to obtain the significant FC matrix across ROIs (p < 0.05 uncorrected). Details of LC functional connectivity analysis are provided in S1 Text.

## EEG effective connectivity analysis: State-space modeling of the latent neural activity

A state-space model was used to infer the latent brain dynamics across the nodes. To infer the activity in the EEG source space, a volumetric source model was used, where EEG sources (denoted as $x_t$ at time $t$) are assumed to be uniformly distributed on a 3D grid inside the brain. Given the observations of the scalp EEG measurements $y_t$, a linear EEG forward model was used: $y_t = Lx_t + e_t$, where $L$ is the lead field matrix and $e_t$ is a Gaussian channel noise vector. We defined the latent state variables as $s_t$, where $x_t = Gs_t + \epsilon_t$, $G$ is a binary indicator matrix, and $\epsilon_t$ is a Gaussian noise vector. The influences across the latent states of neural dynamics were modeled as a multivariate autoregressive (MVAR) process as $s_t = As_{t-1} + \sum_{k=1}^{K} B^k m_t^k s_{t-1} + Du_t + \omega_t$, where $A$ is the intrinsic connectivity matrix, $B^k$ is the $k^{th} (k = 1, 2, \ldots K)$ modulatory connectivity matrix, $m_t^k$ is the modulatory input, $u_t$ is the external input, $D$ is a diagonal matrix denotes the strength of $u_t$, and $\omega_t$ is a Gaussian state noise vector. As for the model inference, the mean-field variational Bayesian approximation was used to make inference on the posterior distributions of the latent space variables and model parameters, during which the evidence lower bound was

maximized. The model is described in detail elsewhere [23]. In this study, we first used the ten salience processing nodes defined from the STV EEG-informed fMRI analysis results. Then, another model was fit with the twelve nodes of SN, DAN, and DMN defined from the HCP-MMP atlas. The ROIs were transformed into each subject's native structural space. The state-space model was fit to each run separately. The modulatory inputs were modeled as a unit height boxcar function with a sequence of events (one each for targets and standards). Here, the ELBO was also used to quantify the quality of the model fitting. For the data of the runs failed in model fitting, we excluded those runs from subsequent analyses. Specifically, 11 out of 90 runs failed in the model fitting with salience processing nodes, and 7 out of 90 runs failed in the model fitting with SN, DAN, and DMN nodes. The estimated posterior distributions of model parameters in the first-level analysis were summarized for group-level Bayesian posterior inference using Bayesian parameter averaging (BPA) [90], which computes a group-level joint posterior probability for the effect of interest. Compared to the conventional statistical null hypothesis significance tests (NHST), the Bayesian posterior inference is considered more robust as the influence from each run is weighted by the precision. The significant group-mean EC was estimated, where a posterior probability criterion of $\alpha < 0.05$ (Bonferroni correction) was used to threshold the group-level posterior distribution for the inference of significant EC.

## Brain and pupil correlation analysis

To assess the relationship between the network-level interactions and pupillary response, we characterized the network-level connectivity in the positive and negative connections. Specifically, we defined positive and negative network interaction strength as the sum of all positive and negative connection parameters from one network node set to the other network node set (or to itself as self-connection network strength), respectively. And we also computed the mean TEPR across all the oddball trials for each subject. Therefore, we performed a correlation analysis across all subjects between the network interaction strength and the mean TEPR (Pearson correlation). As a control analysis, we also regressed out the median RT and the mean ELBO from the mean TEPR and network interaction strength, respectively, before performing the same analysis across subjects.

## Supporting information

**S1 Fig. Stimuli-locked pupillary response.** The z-scored pupil diameter fluctuations from 500 ms before the stimulus to 2000 ms following the stimulus were averaged across subjects for the oddball (red) and standard (yellow) stimuli. The shaded bands represent standard error, and the bottom gray line indicates significant difference (Student's t-test, p < 0.001) between the pupil diameter evoked by the oddball and standard stimuli.
(TIF)

**S2 Fig. Schematic illustration of auditory oddball paradigm, single-trial analysis, and single-trial variability EEG-informed fMRI analysis.** For each temporal window $\tau$, we applied a single-trial analysis with the extracted EEG data in the windows from all the trials, where a logistic regressor was trained to learn a weight matrix $w$ maximally discriminating the target vs standard trials. From the weighting on the EEG channels with matrix $w$, a EEG discriminating component was computed as a low-dimensional representation of the EEG data. For example, two EEG sensors (channel $i$ and $j$) were illustrated in the figure with a hyperplane discriminating target (red dots) and standard (yellow dots) trials. Similarly, single-trial analysis was applied to all other temporal windows spanning the trial independently with a sliding window approach (step size as $\delta$). The EEG discriminating component at each temporal window was

used to modulate regressors in a general linear model (GLM) to predict fMRI BOLD response (convolved with the canonical hemodynamic response function along with other regressors). The GLM analysis was applied with each temporal window independently.
(TIF)

**S3 Fig. Stimulus locked event-related potential at the Pz electrode for the standard (blue) and oddball (red) trials, from 500 ms pre-stimulus to 2000 ms post-stimulus.** The solid lines denote the group mean, and the shaded areas denote the standard error across subjects. The P300 component was observed with a peak around 390 ms.
(TIF)

**S4 Fig. Axial slices of the thresholded group-level significant activations in the traditional fMRI analysis of the oddball effects (contrast as oddball versus standard stimuli).** The z-statistic maps were displayed on top of the MNI152 template brain image. FMRIB's Local Analysis of Mixed Effects (FLAME) from the FSL software package was used for the group-level statistical inference. The group-level statistical parametric maps were thresholded with $z > 3.1$ and corrected cluster significance threshold of $p = 0.05$ (Gaussian random field method). Regions in the dorsal attention network, salience network, visual and auditory cortex, primary somatosensory cortex, and subcortex were identified as significant clusters. Please be noted that only the significant positive effects are shown here, and we did not observe any significant negative effects in the regions of the default mode network. The 'R' in the figure denotes right side of the brain.
(TIF)

**S5 Fig. Group-level mean effective connectivity modulated by the oddball stimuli between salience processing nodes (Bayesian parameter averaging; $\alpha < 0.05$; Bonferroni corrected).** Please be noted the results here reflect mean group effect. The orange and blue color represents positive and negative effective connectivity, respectively.
(TIF)

**S6 Fig. Total connection strength of each salience processing node.** With the effective connectivity results, all the unsigned connection parameters (efferent, afferent and self-connection) associated with the node were summed up to compute the total connection strength. The results suggest that the lSPL and mPFC-SMA have the strongest total connection strength, indicating their roles as hubs in the processing of salience stimuli.
(TIF)

**S7 Fig. Whole brain temporal signal-to-noise ratio (tSNR) analysis was performed to assess the fMRI signal quality especially for the BOLD signal in the LC.** The tSNR was computed for each voxel, by dividing the mean over the standard deviation. (A) Group-level mean tSNR map of preprocessed fMRI data (no spatial smoothing). The tSNR map of each run was spatially normalized into the MNI152 space, and then was averaged across all the runs of subjects. (B) Quantitative analysis and boxplot of tSNR distributions across runs in each ROI. The color denotes the tSNR before (red) and after (blue) the nuisance signal regression (motion parameters and BOLD signals in the 4th ventricle and the left and right hemisphere white matter and lateral ventricles). Before the functional connectivity analysis of the LC, we regressed out the BOLD signal in the 4th ventricle. The tSNR was computed for each voxel in the subject's native functional space, and then was averaged within the ROI (segmented with FreeSurfer). The LC two standard deviation template was used to delineate the LC ROI [91]. The tSNR in the LC is above the standard cut-offs (tSNR > 30) [92].
(TIF)

**S8 Fig. T-statistic maps of the LC seed-based whole-brain functional connectivity results.** We used a mixed effects model for group inference. Each subject's FC map was transformed into z-score with Fisher's Z transformation. And the FC z-score map was thresholded at $p < 0.01$. In the group-level, one sample student's t-test was performed to obtain the significant seed-based FC map of the LC ($p < 0.001$ uncorrected). Significant clusters were identified in the cerebellum, supplementary motor area (SMA), right and left anterior insula (AI), left postcentral gyrus, right precentral gyrus, and thalamus.
(TIF)

**S9 Fig. Network nodes definition with HCP-MMP atlas.** The nodes (circles) of the SN, DMN, and DAN are overlaid with the selected network areas from the HCP-MMP atlas and the MNI152 brain image. The group-level region of interest masks (illustrated as spatial distribution maps) were obtained from majority vote across subjects.
(TIF)

**S10 Fig. Group-level mean effective connectivity modulated by the oddball stimuli between the nodes of DMN, SN and DAN (Bayesian parameter averaging; $\alpha < 0.05$; Bonferroni corrected).** The orange and blue color represents positive and negative effective connectivity, respectively.
(TIF)

**S11 Fig. Flow chart illustrating the steps of data acquisition, preprocessing, single-modality analyses, and cross-modality analyses.**
(TIF)

**S1 Text. Supplementary text.** We provide additional descriptions on data preprocessing, network node definition, locus coeruleus localization, and additional results on brain-pupil relationships.
(PDF)

## Author Contributions

**Conceptualization:** Hengda He, Paul Sajda.

**Data curation:** Hengda He, Linbi Hong.

**Formal analysis:** Hengda He, Linbi Hong.

**Funding acquisition:** Paul Sajda.

**Investigation:** Hengda He, Linbi Hong, Paul Sajda.

**Methodology:** Hengda He, Linbi Hong.

**Project administration:** Paul Sajda.

**Software:** Hengda He, Linbi Hong.

**Supervision:** Paul Sajda.

**Visualization:** Hengda He, Linbi Hong.

**Writing – original draft:** Hengda He.

**Writing – review & editing:** Hengda He, Linbi Hong, Paul Sajda.

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
