## [Decision Letter · Decision Letter 0]

2 Mar 2023

Dear Mr. He,

Thank you very much for submitting your manuscript "Pupillary response is associated with the reset and switching of functional brain networks during salience processing" for consideration at PLOS Computational Biology. As with all papers reviewed by the journal, your manuscript was reviewed by members of the editorial board and by several independent reviewers. The reviewers appreciated the attention to an important topic. Based on the reviews, we are likely to accept this manuscript for publication, providing that you modify the manuscript according to the review recommendations.

Sincerely,

Daniele Marinazzo

Section Editor

PLOS Computational Biology

Reviewer's Responses to Questions

**Comments to the Authors:**

Reviewer #1: This is an interesting and comprehensive multimodal study of the salience network and its relationship to LC. The data and results would be of interest to the readers in the neuroscience community.

Major comment

The data and results presented are quite extensive and interesting, the presentation could be a bit clearer. First, it would be useful to add the chart to illustrate all the steps of the data analyses.

Also, it would be useful to add a summary of the results/conclusion at the end.

It is not clear how the 5 statistical parametric maps obtained with EEG-STV regression analysis of the fMRI are combined and/or used.

Minor comment

I would change "a fMRI" to "an fMRI" on page 5.

Reviewer #2: He et al. report an EEG-fMRI-pupillometry study involving an auditory oddball task in 19 healthy adults. They report a variety of analyses centered on interactions between the LC-NE system and networks including the SN, DAN and DMN. The analyses are rigorous, and the EEG-informed fMRI approach adds important temporal information about network activations during oddball events. The effective connectivity findings provide interesting (though somewhat difficult to interpret) preliminary findings, and the authors acknowledge limitations of that approach. Overall, this a technically impressive multimodal study that has the potential to advance understanding of interactions between neuromodulatory systems (LC-NE) and intrinsic brain networks in relation to cognition. To enhance the impact of the study, my comments center on scholarship, better situating the results within existing literature, clarifications, and some potential inconsistencies with prior findings.

- fMRI research on networks that respond to multimodal (e.g. auditory/visual/somatosensory) salience was conducted well before many of the papers cited by the authors. For example, see Downar et al. (Nat Neurosci 2000; J Neurophysiol 2002) which shows involvement of salience network regions prior to the “salience network” terminology was introduced, using paradigms similar to the one used here. I suggest including these earlier references and discussing in light of the present findings.

- Deactivations in the salience network in response to oddball events were very surprising, given what is usually observed (however, DMN deactivations would be expected). The supplement shows standard GLM results that do seem to show (expected) salience network activations, but the EEG-informed fMRI analyses suggest deactivations. What explains this discrepancy, and were DMN deactivations found in the standard GLM analysis (only activations are shown)?

- The results in figure 5 are based on individual differences and the sample size is small, raising questions about power and statistical bias. To support these findings, could the authors look at intra-individual, trial-by-trial relationships (which would be well powered and would still address the question of interest here)?

- The final section of Results focus on the LC, but the second half of that section focused only on pupillary responses (TEPR) as an index of the LC. The authors did examine LC activity for functional connectivity analyses but it does not appear that they looked at LC phasic responses and their relationships with pupillary responses and cortical networks. Rather than using TEPR as an index of LC here, why not look at LC activity itself?

- Several intracranial EEG studies have been mapping out the temporal dynamics of task-evoked responses in DAN, SN, and DMN (Raccah et al 2018 J Neurosci; Kucyi et al 2020 Nat Commun) as well as relationships between the SN and pupillary responses (Kucyi et al 2020 J Neurosci). Such findings also highlight early DAN responses (prior to SN) and SN-pupil relationships potentially implicating the LC-NE system. The discussions in this paper, while rigorous in terms of fMRI and EEG/MEG literature, have not included these very (likely even more) relevant references that relate strongly to the present findings. How do these findings build on such prior intracranial EEG work?

- The discussion section on broader implications of pupillometry-EEG-fMRI could better acknowledge prior studies that have also used this valuable approach (e.g. Groot et al., 2021 Neuroimage).

- More data acquisition specifications for EEG conducted with fMRI are needed in the Methods section. For example, was the MR clock synchronized to EEG recordings to reduce noise? How were the electrode impedance levels set and tested prior to subjects entering the scanner?

**Have the authors made all data and (if applicable) computational code underlying the findings in their manuscript fully available?**

Reviewer #1: Yes

Reviewer #2: Yes

PLOS authors have the option to publish the peer review history of their article (what does this mean?). If published, this will include your full peer review and any attached files.

Reviewer #1: No

Reviewer #2: No

Figure Files:

Data Requirements:

Reproducibility:

References:

---

## [Editor Report · Decision Letter 1]

6 Apr 2023

Dear Mr. He,

We are pleased to inform you that your manuscript 'Pupillary response is associated with the reset and switching of functional brain networks during salience processing' has been provisionally accepted for publication in PLOS Computational Biology.

Best regards,

Daniele Marinazzo

Section Editor

PLOS Computational Biology

Daniele Marinazzo

Section Editor

PLOS Computational Biology

---

## [Editor Report · Acceptance letter]

9 May 2023

PCOMPBIOL-D-22-01699R1 

Pupillary response is associated with the reset and switching of functional brain networks during salience processing

Dear Dr He,

I am pleased to inform you that your manuscript has been formally accepted for publication in PLOS Computational Biology. Your manuscript is now with our production department and you will be notified of the publication date in due course.

With kind regards,

Bernadett Koltai
